# LARGE CONTENT AND BEHAVIOR MODELS TO UNDERSTAND, SIMULATE, AND OPTIMIZE CONTENT AND BEHAVIOR

**Ashmit Khandelwal**[*,1]    **Aditya Agrawal**[*,1]   **Aanisha Bhattacharyya**[*]   **Yaman K Singla**[*]

**Somesh Singh**   **Uttaran Bhattacharya**   **Ishita Dasgupta**   **Stefano Petrangeli**

**Rajiv Ratn Shah**   **Changyou Chen**   **Balaji Krishnamurthy**

Adobe,   BITS, Pilani,   IIIT-Delhi,   State University of New York at Buffalo

## ABSTRACT

Shannon and Weaver's seminal information theory divides communication into three levels: *technical*, *semantic*, and *effectiveness*. While the technical level deals with the accurate reconstruction of transmitted symbols, the semantic and effectiveness levels deal with the inferred meaning and its effect on the receiver. Large Language Models (LLMs), with their wide generalizability, make some progress towards the second level. However, LLMs and other communication models are not conventionally designed for predicting and optimizing communication for desired receiver behaviors and intents. As a result, the *effectiveness* level remains largely untouched by modern communication systems. In this paper, we introduce the receivers' "behavior tokens," such as shares, likes, clicks, purchases, and retweets, in the LLM's training corpora to optimize content for the receivers and predict their behaviors. Other than showing similar performance to LLMs on content understanding tasks, our trained models show generalization capabilities on the behavior dimension for behavior simulation, content simulation, behavior understanding, and behavior domain adaptation. We show results on all these capabilities using a wide range of tasks on three corpora. We call these models Large Content and Behavior Models (LCBMs). Further, to spur more research on LCBMs, we release our new Content Behavior Corpus (CBC), a repository containing communicator, message, and corresponding receiver behavior[*].

## 1 INTRODUCTION

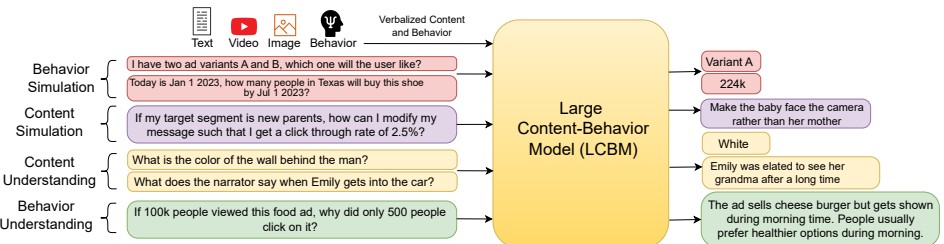

Figure 1: Encoding and predicting content (images, videos, and text) and behavior in the language space. Large Content Behavior Models (LCBMs), once trained, can enable a host of different applications, including behavior simulation, content understanding, content-behavior optimization, and content-behavior understanding.

---

[*]Equal Contribution. Contact ykumar@adobe.com for questions and suggestions.

[1] Work done while at Adobe.

[*]https://behavior-in-the-wild.github.io/LCBM

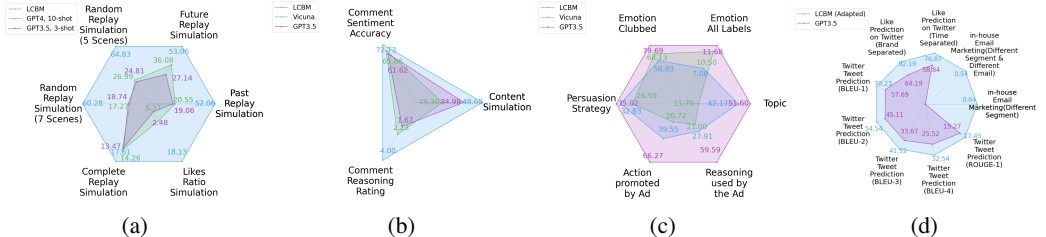

Figure 2: Comparison of GPT-3.5, GPT-4, Vicuna-13B, and LCBM-13B on: (a) Behavior Simulation accuracy on two types of behaviors: replay value prediction and likes/views prediction. The task is, given the video content and channel information, to predict replay values corresponding to each scene and the ratio of likes to views. (b) Content simulation and behavior understanding tasks. The task for content simulation is, given the channel information and scene-level behavior, to predict the scene content. Given information on the video platform and the video content, the task of behavior understanding is to predict and explain the sentiments of the viewers and the commenters. Six evaluators scored the models' explanations between 0-5 to get the predicted sentiment and explanation scores by comparing the ratings and reasons with the user comments. The annotators did not know which model gave the reasoning. (c) Content understanding tasks. We evaluate four tasks: emotion, topic, and persuasion strategy prediction, and action-and-reason understanding. (d) Behavior Simulation on in-house Email Marketing Data ($R^2$ score) and Twitter likes (accuracy), and Content Simulation on Twitter tweet prediction (BLEU/ROUGE scores). It can be noted that on the behavior simulation, content simulation, and behavior understanding tasks, LCBM performs better than 3-shot GPT-3.5 and 10-shot GPT-4 (covering a larger area. On the content understanding tasks, while LCBM outperforms similar-sized Vicuna models, GPT-3.5 performs better. However, we also note that GPT-3.5 and GPT-4 are at least 12 times larger than LCBM-13B. Further, we show the behavior domain adaptation results in Table 6, 7, 8.

Shannon & Weaver (1949), in their seminal paper on communication, includes all of the procedures by which one mind may affect another. This includes all forms of expression, such as words, gestures, speech, pictures, and musical sounds. They mentioned that the broad problem of communication can be studied at three levels: technical, semantic, and effectiveness.

**Level A: Technical.** How accurately can the symbols of communication be transmitted?

**Level B: Semantic.** How precisely do the transmitted symbols convey the desired meaning?

**Level C: Effectiveness.** How well does the received meaning induce the desired conduct in the receiver?

These three levels build on top of each other. Thus, solving the problem at Level C necessarily requires solving the corresponding problems at Levels A and B.

Since the publication of this seminal paper, the tremendous growth in the field of telecommunications, particularly the advent of the Internet and mobile devices, has led to affordable, wide-scale solutions for Level A. With the recent advances in large language models (LLMs) such as BERT (Devlin et al., 2018), GPT-3 and 4 (Brown et al., 2020; OpenAI, 2023), T5 (Raffel et al., 2020), and many more, we have witnessed a significant improvement in the performance of various Natural Language Processing (NLP) tasks. LLMs in zero- or few-shot settings can easily handle tasks such as question answering, summarization, translation, and many more. This has helped us progress towards solving Level B to a large extent. However, the Level C problem of effectiveness remains largely unsolved. Effectiveness refers to designing messages that can fulfill the communicators' underlying objectives, such as explaining complex concepts to the receivers and informing the receivers' choices (*e.g.*, when making purchase decisions).

**How do we solve the effectiveness problem while retaining the other two levels?** To solve the effectiveness problem, we can take inspiration from how the semantic problem is being solved. Raffel et al. (2020), in their seminal work on T5, mention that the basic idea underlying large language models is to treat every text processing problem as a "text-to-text" problem, *i.e.*, taking the text as input and producing new text as output. This framework allows for a direct application of the same model, objective, training procedure, and decoding process to every task we consider. Further, this allows us to pre-train a model on a data-rich task like the next-word prediction, which can then be transferred to downstream tasks. Notably, thanks to the Internet, the next-word prediction task has huge amounts of available data. Consider the Common Crawl project (https://commoncrawl.org),

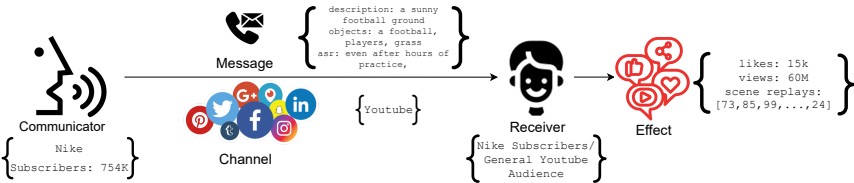

Figure 3: Five factors of communication: Communicator, Message, Channel, Receiver, and Effect.

one common source of data included in most language models. It produces more than 20TB of text per month sampled from random web pages across the internet.

T5 and other language models like GPT-3, Pythia (Biderman et al., 2023), and Llama (Touvron et al., 2023) can solve a wide variety of tasks, including the ones for which they were not explicitly trained. For instance, language models trained on the next word prediction task showed generalization capabilities across a wide variety of tasks like question-answering, summarization, natural language inference, and translation (Brown et al., 2020). Recently, a series of papers have shown that this generalized "world understanding" captured in LLMs can be leveraged to enable them to "see" (Liu et al., 2023; Li et al., 2023b;a; Zhu et al., 2023; Ge et al., 2023; Zhang et al., 2023; Bhattacharyya et al., 2023). This is a significant capability enhancement since a model trained in language only settings can be made to reason about images and videos. These papers follow the same transfer learning approach advocated by T5, where they convert visual information to language space to leverage the "text-to-text" framework. They show that it is possible to teach a large language model, the new modality of vision, without needing to pre-train the model from scratch. Rather, using only a few million tokens, it is possible to scale LLMs' abilities to vision as well. Following this chain of thought, it could be possible to solve the effectiveness problem by posing it as a "text-to-text" problem. This is the paradigm we explore in this work.

**How can we pose the effectiveness problem as a text-to-text problem?** The problem of effect is to know what the receiver does after receiving the message (Shannon & Weaver, 1949). In general, for a piece of content, other than the content itself, we often have information about *who* consumes the content and what his *action* is on consuming the content. The latter is the effect described in Shannon's three levels of communication. For instance, an email, as a message from the communicator to the receiver, elicits certain actions from the receiver like link-clicks, replies, and read-time. While LLMs are trained on trillions of tokens of content, the training does not include the receiver effect. For instance, Enron Email (Klimt & Yang, 2004) is a popular corpus that is included in the training of LLMs like Pythia (Biderman et al., 2023). It contains 600K email content sourced from the Enron corporation, which LLMs use to learn how to write emails. However, it does not contain data about the receivers' activities, such as whether they opened the email, how long they kept it open (read-time), and what their reply was. Similarly, while major text corpora include a large number of public blogs and user forums to train LLMs like CommonCrawl, they are stripped of receiver behavior on forum messages, such as the number of likes, shares, and replies, before including them in LLM training (for instance, see (Biderman et al., 2022; Penedo et al., 2023)). To pose the effectiveness problem as a text-to-text problem, we can include these *behavior tokens* in the text along with content tokens and train the LLM to model both of those in the same space. This might help the LLM simulate the receiver effect, optimize for it, and reason about it.

In this paper, we show initial experiments to integrate behavior as a new modality to increase the scope of multimodal LLMs from only content to both content and behavior. We call this new type of model a Large Content Behavior Model (LCBM). This class of models shows promise in enabling the LLMs to not only reason about content but also reason about and predict human behavior over that content. Further, LCBMs have the potential for behavior domain adaptation where models trained on one type of behavior can generalize on another behavior type (Fig. 2). Behavior simulation can enable many real-world applications, such as content recommendation, customer journey optimization, and A/B testing. To build LCBM, we introduce behavior instruction tuning (§2.4), an attempt to extend the instruction tuning paradigm to behavior space, bringing all five communication factors (communicator, message, channel, receiver, and effect) into the same space (Fig. 3). Similar to Brown et al. (2020); Raffel et al. (2020); Liu et al. (2023); Ge et al. (2023), we do not

design best-in-class predictors for any of the downstream tasks. Rather, we show a model which shows generalization capabilities across a wide variety of content- and behavior-related tasks. To summarize, our paper makes the following two contributions:

- **Large Content Behavior Model (LCBM).** We develop a large multimodal model that shows capabilities of behavior simulation (given content), content simulation (given behavior), content understanding, and behavior understanding (Fig. 1). Following the text-to-text framework, we connect the Vicuna LLM (Touvron et al., 2023; Chiang et al., 2023) with an open-set visual encoder of EVA-CLIP (Sun et al., 2023) and instruction fine-tune it end-to-end on behavior instruction data. EVA-CLIP and QFormer (Li et al., 2023a) help the model to understand visual content in the language space, making it a Vision Language Model (VLM). During behavior instruction tuning, we teach the model to predict behavior given content and content given behavior using various instruction tasks (§2.4). This helps us teach behavior modality to the VLM while grounding it in the natural language space. We use three datasets to show the performance of LCBM: a dataset consisting of YouTube videos as the content and the corresponding retention graph, likes, the number of views, and comment sentiment as receiver behavior; a dataset consisting of Twitter posts (text, images, and videos) and corresponding human behavior (like counts) extracted from 168 million tweets across 10135 enterprise Twitter accounts from 2007 to 2023 Khurana et al. (2023b); and an internal dataset of In-house Marketing Emails[‖] (content) and the click-through rate corresponding to each segment they were sent to (behavior). We observe that teaching the LCBM behavior and content simulation improves its capabilities on them (expected), but the model also shows signs of domain-adaptation in behavior modality (few-shot capability, *unexpected*) (Tables 6,7,8) and improvements in behavior understanding (Figs. 5,6,§3) (zero-shot capability, *unexpected*) (Brown et al., 2020). See Fig. 2 for a radar plot of all the capabilities and comparisons of performances across LCBM and state-of-the-art LLMs: GPT-3.5 and GPT-4.
- **Dataset and Test Benchmark.** To spur research on the topic of large content and behavior models, we release our generated behavior instruction fine-tuning data from over 40,000 public-domain YouTube videos and 168 million Twitter posts. The data contains: 1) YouTube video links, automatically extracted key scenes, scene verbalizations, replay graph data, video views, likes, comments, channel name, and subscriber count at the time of collection, and 2) Twitter extracted account names, tweet text, associated media (image and video) verbalizations (including image captions, keywords, colors, and tones), tweet timestamps, and like counts Khurana et al. (2023b). We also release a benchmark to test performance on the joint content behavior space (§2.3), introducing two types of tasks in this space: predictive and descriptive. In the predictive benchmark, we test the model's ability to predict behavior given the content and predict content given the behavior. In the descriptive benchmark, we validate its explanation of human behavior by comparing it with ground-truth annotations we obtain from human annotators that try to explain human behavior. See Figs. 5,6 for a few examples.

## 2 SETUP

In this section, we introduce our approach to model content and behavior together as a text-to-text problem. Since most publicly available corpora strip off receiver behavior from content, we first introduce our dataset, "The Content Behavior Corpus (CBC)", a dataset consisting of content and the corresponding receiver behavior. Next, we introduce our methodology to convert the content and behavior into text and our approach to model it using an LLM. Then, we cover the tasks through which we test various capabilities of LCBM (Fig. 1): content-understanding, behavior understanding, content simulation, behavior simulation, and behavior domain adaptation.

### 2.1 THE CONTENT BEHAVIOR CORPUS (CBC)

The availability of large-scale unlabeled text data for unsupervised learning has fueled much of the progress of LLMs. In this paper, we are interested in modeling content and the corresponding receiver behavior in the same space. While available datasets contain trillions of content tokens (text, images, audio, and videos), they unfortunately do not contain the receiver effect. To address this, we utilize YouTube and Twitter, two large publicly available sources of content-behavior data,

---

[‖]We obtain In-house Marketing Emails dataset by collaborating with the In-house team.

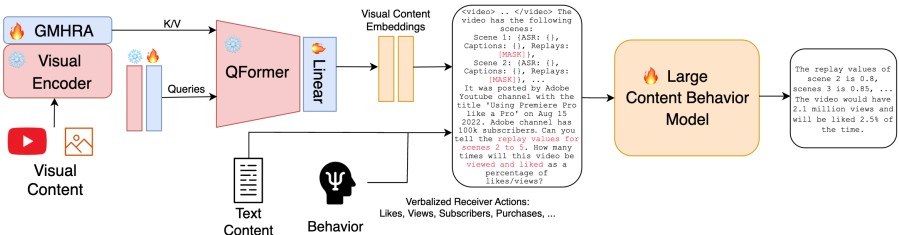

Figure 4: Encoding and predicting content (images, videos, and text) and behavior in the language space. Strategy to behavior instruction fine-tune (BFT) LLMs to create LCBMs. We capture visual concepts through the visual encoder (EVA-CLIP), and world knowledge is through an LLM (Llama). To leverage the rich knowledge of LLMs, we use GMHRA and QFormer to convert visual tokens of ViT to language tokens that Llama can understand. Further, we find that verbalizing the visual stimulus helps Llama to gather information more explicitly than what is provided by ViT+QFormer. We fine-tune the combined model end-to-end to predict 1) behavior given content and 2) content given behavior. Snowflake and fire symbols denote the frozen and unfrozen parts of the architecture.

consisting of (a) account name, account description, and number of subscribers and followers (*communicator data*), (b) rich content in the form of videos, images, creator-provided captions, titles, and descriptions (*message*), (c) behavior in the form of likes, views, user comments, and replay graph (*receiver effect*). This covers all the five factors of communication (Fig. 3), with the channel being fixed (as YouTube or Twitter) and receivers being average channel followers and viewers of the communicator. Since content data is multimodal in the form of a combination of images, videos, and text, and behavior data is in the form of numbers, to model it using a text-to-text paradigm, we *verbalize* both of them following the methodology we detail next.

*Verbalization:* For the video $V$, YouTube provides us with 100 average viewer retention values $r_i$ for $i \in [0..100)$, corresponding to the entire video. The sampling rate of 100 is constant and independent of video length ($T$). Replay value $r_i$ corresponds to video frames between the timestamps $(T/100 \times i, T/100 \times (i+1))$, which denotes how often these frames were replayed compared to the most replayed frames. The metric has a value between 0 and 1 that identifies the video's relative retention performance at a given point in the video. To accommodate longer video lengths, we merge replay values until $T/100 \times (i+j) - T/100 \times i > 1$ second with $j \in \{i+1, 100\}$. We choose the replay value for this merged group of scenes as $max(r_i, ..., r_j)$. Using this logic, we get replay values $R_i$ for $i \in [0..m]$, where $m = \lfloor 100/(\lceil 100/T \rceil) \rfloor$. Next, we sample two frames randomly corresponding to each $i \in [0..m]$. We caption the frames using BLIP (Li et al., 2023a). We also obtain the automatic speech recognition for the speech for the video between the timestamps corresponding to replay value $R_i$ using Whisper (Radford et al., 2023). The ASR and BLIP captions are content for scenes, and replay values are the behavior corresponding to them. We include the scene content and behavior in the video verbalization (Listing 1) with the sampling for both scene content and behavior as described above.

We also include video content by encoding video frames through EVA-CLIP (Sun et al., 2023) (explained in §2.2). Other than video embeddings, we include the video title and description as part of the video content. Corresponding to the overall video content, we verbalize overall video behavior metrics like video views and the ratio of likes and views. Finally, we append it with communicator information on the video channel and the subscriber count. The Listing 1 presents the overall verbalization for video and frame level content and behavior. The verbalization for Twitter posts is similar and is given in Listing 7.

Listing 1: Verbalization pattern for inputting content and behavior in the same space

```
Input: <video> ..[Video Tokens] .. </video>
The video has the following scenes:
Scene 1: {ASR: Welcome to a quick tutorial, OCR: Adobe Premiere Pro, Captions: A desktop interface, Replays: 60},
Scene 2: {ASR: on using Premiere Pro to edit, Captions: A computer interface, with an image of a white horse. Objects – Horse,
    Grass, Fence., Replays: 53},
...
It was posted on Adobe's YouTube channel with the title 'Using Premiere Pro like a Pro' on Aug 15 2022. Adobe's YouTube
    channel has 100k subscribers. This video was viewed by 346 thousand people and liked (as a percentage of likes/views) by
    2.3% people.
```

| Model | #Params | Training | Past | | Future | | Random Window Size | | | | All Masked | |
|---|---|---|---|---|---|---|---|---|---|---|---|---|
| | | | | | | | 5 | | 7 | | | |
| | | | RMSE | Accuracy | RMSE | Accuracy | RMSE | Accuracy | RMSE | Accuracy | RMSE | Accuracy |
| LCBM | | 3-BFT | 8.12 | 55.10 | 15.05 | 42.42 | 8.55 | 61.41 | 9.91 | 55.10 | - | - |
| LCBM | 13B | 5-BFT | 11.53 | 52.06 | 12.02 | 53.06 | 8.13 | 64.83 | 9.22 | 60.26 | 31.34 | 17.16 |
| LCBM | | 7-BFT | 16.17 | 35.61 | 15.14 | 44.11 | 9.02 | 59.22 | 10.47 | 53.84 | - | - |
| LCBM | | 11-BFT | 18.25 | 30.95 | 15.05 | 41.44 | 10.01 | 55.15 | 10.49 | 52.61 | - | - |
| GPT-4 | >100B[†] | 10-shot-ICL | 34.45 | 20.55 | 19.51 | 36.08 | 22.99 | 26.99 | 27.25 | 17.27 | 38.52 | 14.26 |
| GPT-4 | | 2-shot-ICL | 35.05 | 19.34 | 18.07 | 39.33 | 17.42 | 38.10 | 21.26 | 28.05 | 37.60 | 13.73 |
| GPT-3.5 | 175B | 3-shot-ICL | 34.10 | 19.06 | 24.71 | 27.14 | 24.52 | 24.81 | 26.30 | 18.74 | 38.77 | 13.47 |
| GPT-3.5 | | 2-shot-ICL | 33.36 | 18.02 | 26.44 | 25.42 | 23.35 | 25.35 | 24.68 | 21.24 | 37.16 | 13.39 |
| Random | - | - | 34.10 | 10.00 | 34.10 | 10.00 | 34.10 | 10.00 | 34.10 | 10.00 | 34.10 | 10.00 |

Table 1: **Behavior Simulation.** Mean RMSE and accuracy scores for scene-by-scene predictions of video replay values. Replay values are the normalized replay scores of each scene as provided by YouTube. The normalized scores are considered to 2 decimal places and multiplied by hundred to convert the score to an integer score in the range 0-100. RMSE is calculated for each video in the test set and the mean is calculated for this score and reported. The model is said to classify correctly if the absolute error between the predicted and ground truth value is less than or equal to 5. The scores are calculated in four regimes: past, future, random, and all-masked. In the past (future) regimes, first (last) 5-20% scenes are masked; in the random setting, 5-20% scenes are masked randomly, and in all masked setting, everything is masked. LCBM was behavior-fine-tuned (BFT) with 3,5,7,11 context window masking strategy, while GPT was compared with an in-context learning (ICL) setting. We note that behavior fine-tuned LCBM, while being at least 10x smaller than other models, performs the best. Best models are denoted in green and runner-ups in blue.

## 2.2 MODEL

To understand both visual and textual contents, we follow a similar approach as was taken by recent models like BLIP, Llava, VideoLlama, and others (Liu et al., 2023; Ge et al., 2023; Li et al., 2023a; Zhu et al., 2023), we use visual encoders to encode visual knowledge and an LLM to encode text and world knowledge. Fig. 4 shows our architecture to encode visual content into the language space. We include video content by encoding video frames through EVA-CLIP (Sun et al., 2023) and Global Multi-Head Relation Aggregator (GMHRA) from Uniformer (Li et al., 2021). GMHRA helps aggregate the information better across the time dimension. The combination of ViT and GMHRA gives us a good representation of the visual content. Next, to effectively leverage the LLM's rich language representations, we use Q-Former from BLIP-2 (Li et al., 2023a) with an extra linear layer and additional query tokens to convert from visual tokens to language tokens. Further, similar to Bhattacharyya et al. (2023), we find that while encoding visual tokens is powerful, converting visual content to text adds to the downstream performance. Therefore, we include the BLIP caption for each scene along with the scene replay graph.

We use the Llama-based Vicuna-13B LLM (Touvron et al., 2023; Chiang et al., 2023) as our base LLM. Similar to prior works (Liu et al., 2023; Ge et al., 2023; Li et al., 2023a; Zhu et al., 2023), we follow a two-stage training paradigm where in the first stage, we utilize the WebVid (Bain et al., 2021), COCO caption (Chen et al., 2015), Visual Genome (Krishna et al., 2017), CC3M (Sharma et al., 2018), and CC12M (Changpinyo et al., 2021) datasets to align the visual encoder embeddings with LLM. In the second stage, we train the model with behavior instructions prepared by following the approach described in §2.4. In summary, LCBM takes concatenated inputs of visual tokens, scene ASR, caption, scene behavior of replays, channel information, and video title and behavior metrics of views and a ratio of likes to views. Based on the instruction, we test LCBM's abilities on various tasks we cover in the next paragraphs.

## 2.3 CONTENT BEHAVIOR TEST BENCHMARK

We test the capabilities of large content-behavior models on predictive and descriptive abilities on content and behavior, as illustrated in Fig: 1. We design the following five tasks to test these capabilities: behavior simulation, content simulation, content understanding, behavior understanding, and behavior domain adaptation. We cover each of these tasks next.

1. **Behavior Simulation.** We test simulation capability on four behaviors across two datasets: YouTube replay values, the ratio of YouTube likes to views, Twitter likes, and the number of views of the YouTube video. The common task amongst all of them is to predict the behavior given the content and content attributes like captions, scene-by-scene descriptions for videos, and sender characteristics like account and subscriber count and date of posting. The behavior to be predicted is masked and asked as a question to the LLM. Listings 6 and 7 lists the verbalization pattern for this task. For replay value prediction, we test the masked behavior in three settings: *Masked Past* (all replay values of the first 5-20% scenes are masked), *Masked Future* (all replay values of last 5-20% scenes are masked), and *Random Masks* (random masking of replay values for 5-20% scenes).

2. **Content Simulation.** Here, the task is to predict content given receiver behavior (Listing 5, 8). For YouTube, given the video content in terms of scene-by-scene descriptions with the content of one group of five consecutive scenes content being masked, behavior values of all scenes, and channel information, the task is to choose the masked scene speech from a list of 25 options, chosen randomly from the entire test set. For YouTube, we chose to model this task as a discriminative task instead of a generative one since videos are generally long, and there could be multiple possible contents for a given behavior, whereas the ground truth is available only for one specific characterization of the content for a given behavior. For Twitter, we model this task as content generation. The Listing 8 presents the format for this task.

3. **Behavior Understanding.** The goal of this task is to check if the model can reason about observed or unobserved receiver behavior. For this task, we could ask the model to explain any behaviors given the content. However, only the YouTube receiver comments have ground truth available with the video. Without ground truth, we found that other behaviors, such as replay values, likes, and views, are difficult to explain by non-experts. Therefore, we ask the model to simulate the sentiment of the receivers' comments and describe its reasoning. To evaluate, we asked six annotators to annotate the reasons provided by the model on a scale of 0-5, with 0 implying the LLMs provided no sentiment or reasoning and 5 implying perfect reasoning. The annotators were free to rate the LLMs as they seemed fit. The annotators were asked to review the video content and the comments to help them evaluate the reasons. We average the ratings of three annotators to get an average rating for every video. Similarly, to review the sentiment correctness, we asked the annotators to judge the predicted sentiment rating with respect to user comments.

4. **Content Understanding.** To check if a model trained on both content and behavior tokens does not forget its original content understanding capabilities, we test the content understanding tasks on YouTube videos, following Bhattacharyya et al. (2023). They use the following tasks for video-understanding: topic, emotion, persuasion, and action-reason classification. For topic, emotion, and action-reason classification tasks, they use the advertisements dataset by Hussain et al. (2017), which contains 3,477 video advertisements and the corresponding annotations for emotion and topic tags and action-reason statements for each video. There are a total of 38 topics and 30 unique emotion tags per video. Further, we have 5 action-reason statements for each video for the action-reason generation task. For our experiment, we use the subset of 1,785 public videos. Following Bhattacharyya et al. (2023), for the topic and emotion classification task, we evaluate our pipeline using top-1 accuracy as the evaluation metric. Further, we evaluate emotion classification on clubbed emotion labels as well. For action and reason prediction, we evaluate our accuracy on the action and reason retrieval tasks where 29 random options along with 1 ground truth are provided to the model to find which one is the ground truth. In the persuasion strategy classification, we use the 1002 persuasion strategy videos and corresponding labels released by Bhattacharyya et al. (2023). Given the video, the model has to predict which persuasion strategy the video conveys. Persuasion strategy classification could be an important task for evaluating LCBM since the concept of persuasion in psychology views human communication as the means to change the receiver's beliefs and actions (*i.e.*, to persuade) (Kumar et al., 2023), and understanding the different strategies present in communication may help understand human behavior better. We evaluate the top-1 accuracy of the model on this task.

5. **Behavior Domain Adaptation.** In the past work, we have observed strong generalization capabilities from LLMs (OpenAI, 2023; Ouyang et al., 2022; Raffel et al., 2020). While training on next token prediction, LLMs show generalization across tasks, including question answering, natural language inference, and sentiment analysis. Given this, the natural question is, does LCBM, too, show this kind of generalization, where a model trained on one kind of behavior, can show performance on another behavior? To understand this, we test the model on a different

dataset and task than what it was originally trained for. We do this over three datasets, LVU (Wu & Krahenbuhl, 2021), In-house Email Marketing[‖], and generalization between Twitter and YouTube likes.

- **LVU Benchmark.** Wu & Krahenbuhl (2021) released a benchmark for long video understanding with over 1000 hours of video. In the benchmark, they have two behavior related tasks: ratio of likes to likes+dislikes and view prediction. YouTube has discontinued the dislike count, therefore, our corpus does not contain the dislike count. We use the LVU test benchmark to check if a model trained on other available behaviors (views, likes, and replay graphs) is able to predict the like ratio.
- **In-house Email Marketing.** In this task, we ask the model to predict the click-through rate for a given target segment of an email, given the email content, subject, and verbalized descriptions of the images in the email. We use the emails sent by In-house marketing team to its subscribers. The emails were sent from April 1, 2022 to June 12, 2023 and covered many of the premiere products The emails were sent to many customer segments (as defined by the marketing team) across 225 countries (Table 9). Listing 3 lists the verbalization format to verbalize emails to input to the LCBM.

## 2.4 BEHAVIOR INSTRUCTION FINE-TUNING (BFT)

To teach an LLM the behavior modality over multimodal content, we convert both the visual tokens and behavior modality in the text format and instruction fine-tune the LLM end to end. This follows a two-stage approach: first, we teach the LLM the visual modality (§2.2), and next, we teach the LLM the behavior modality. We call the latter "Behavior Instruction Fine-Tuning (BFT)" inspired by instruction fine-tuning (IFT) and its variants like visual instruction tuning (Liu et al., 2023).

We prepare the content-behavior instruction datasets as explained next.
**Teaching behavior in the forward direction** (predict behavior given content): In this instruction tuning task, we teach the model to predict behavior given the message sent by the communicator. Essentially, this teaches the model to predict behavior in the forward direction (as in Fig. 3). Concretely, we include the following information as part of verbalization - image and video embedding converted to the text space (using EvaCLiP (Sun et al., 2023)), scene-by-scene verbalization covering automatic speech recognition, scene captions, video/post caption and description, receiver behavior covering replay rates, views, and likes, and communicator information covering account name and follower count. The verbalisation pattern for this task is the same as given in the Listing 6.

**Teaching behavior in the reverse direction** (predict content given behavior): This task teaches the model to learn about behavior in the reverse direction (Fig. 3). Here, the model learns to simulate content given behavior. The instruction for this task is given in Listing 4.

Using the prepared content and behavior instruction datasets consisting of pairs of content and behavior tokens, we treat the content tokens ($\mathbf{X}_C$) as input and behavior tokens ($\mathbf{X}_B, x_i \in \mathbf{X}_B$) as output of the language model. We then perform instruction-tuning of the LLM on the prediction tokens, using its original auto-regressive training objective. Specifically, for a sequence of length L, we compute the probability of generating target answers ($\mathbf{X}_B$) by:

$$p(\mathbf{X}_B|\mathbf{X}_C) = \prod_{i=1}^{L} p_\theta(x_i|\mathbf{X}_C, \mathbf{X}_{B,<i}) \tag{1}$$

For the behavior instruction tuning, we keep the visual encoder weights frozen and continue to update the pre-trained weights of the LLM in LCBM.

## 3 RESULTS AND DISCUSSION

Here, we discuss the results for the five tasks we discuss in Section 2.3, namely, behavior simulation, content simulation, behavior understanding, content understanding, and behavior domain adaptation. We compare the behavior fine-tuned model discussed in §2.4 with state-of-the-art content-only models like GPT-3.5, GPT-4, and Vicuna-13B. This allows us to compare how much including behavior tokens in the training of an LLM helps in improving the LLM's understanding of behavior and joint content and behavior spaces while retaining its understanding of the content space.

| Model | #Params | Accuracy |
|-------|---------|----------|
| Vicuna | 13B | 19.30% |
| LCBM | 13B | 48.68% |
| GPT-3.5 | 175B | 34.98% |
| Random | - | 4% |

Table 2: **Content Simulation.** In this task, the models have to choose the speech segment from a list of 25 options given the video description, non-masked scenes. and replay behavior. We see that despite being similar to masked language modeling (which is a content-only task), LCBM performs better than both Vicuna and GPT-3.5. Best models are denoted in green and runner-ups in blue .

| Model | #Params | Sentiment Accuracy | Reasoning Score |
|-------|---------|--------------------|-----------------|
| **Vicuna** | 13B | 65.66% | 2.23 |
| **LCBM** | 13B | 72.73% | 4.00 |
| **GPT-3.5** | 175B | 61.62% | 1.67 |

Table 3: **Behavior Understanding.** In this task, the models have to simulate the sentiment of comments that a video would get by looking at only the video. Further, they also have to explain the reason for such sentiment. The responses were annotated by humans on a scale of 0-5 for the reason, with 0 being no response provided and 5 being the response matches exactly with the (ground truth) comments received on the video. Best models are denoted in green and runner-ups in blue .

The results for the five tasks are presented in Tables 1,4,2,3,5, 6,7, and 8. We note a few general trends. LCBM, while being 10x smaller than GPT-3.5 and 4, performs better than them on all behavior-related tasks. Further, we see that there is no significant difference between 10-shot and 2-shot GPT-4 or between GPT-3.5 and GPT-4, indicating that unlike other tasks, it is harder to achieve good performance through in-context-learning on the behavior modality. It can be observed that often GPT-3.5 and 4 achieve performance comparable to (or worse than) random baselines. Interestingly, the performance of GPTs on the content simulation task is also substantially behind LCBM. The way we formulate the content simulation task (Listing 5), it can be seen that a substantial performance could be achieved by strong content knowledge, and behavior brings in little variance. We still see a substantial performance gap between the two models. All of this indicates that large models like GPT-3.5 and 4 are not trained on behavior tokens.

For the content understanding tasks (Table 5), predictably GPT-3.5, being the largest model, achieves the best results. However, we see that BFT helps the LLM to learn most content understanding tasks better than the base LLM. LCBM gets better results than both Vicuna and VideoChat. This indicates that behavior modality might carry additional information about the content, which might help an LLM understand content better (Khurana et al., 2023a; Klerke et al., 2016; Plank, 2016). Next, we see that LCBM also shows signs of domain adaptation in the behavior modality. We see that on five tasks: comment sentiment prediction, comment sentiment reasoning (Table 3), email behavior simulation (Table 6), and Twitter behavior (Table 7) and content simulation (Table 8). We note that if the LCBM is trained on only email behavior simulation samples, it underperforms the model trained on both YouTube data and a few samples to make the model learn email format. Similarly, LCBM trained on both Twitter and YouTube performs better than the one just trained on Twitter, showing performance improvement by domain adaptation. Finally, Figs. 5,6 show a few samples where we query LCBM to explain replay and comment behavior and compare it with human explanations. We see that LCBM while verbose, can explain behavior well.

## 4 CONCLUSION

In this paper, we make initial strides towards solving the effectiveness problem proposed by Shannon in his seminal paper on communication. The effectiveness problem deals with predicting and optimizing communication to get the desired receiver behavior. This can be seen as consisting of a string of capabilities: behavior simulation, content simulation, and behavior domain adaptation. We show that while large language models have great generalization capabilities, are unable to perform well on the effectiveness problem. We posit that the reason for this could be a lack of "behavior tokens" in their training corpora. Next, we train LLMs on behavior tokens to show that other than content understanding tasks, the trained models are now able to have good performance across all the behavior-related tasks as well. We also introduce a new Content Behavior Corpus (CBC) to spur research on these large content and behavior models (LCBMs).

---

§Note that we cannot compare this model with GPT-3 due to the private nature of data.

ACKNOWLEDGEMENTS

Rajiv Ratn Shah is partly supported by the Infosys Center for AI, the Center of Design and New Media, and the Center of Excellence in Healthcare at IIIT Delhi.

Changyou Chen is partially supported by NSF AI Institute-2229873, NSF RI-2223292, an Amazon research award, and an Adobe gift fund. Any opinions, findings, conclusions, or recommendations expressed in this material are those of the author(s) and do not necessarily reflect the views of the National Science Foundation, the Institute of Education Sciences, or the U.S. Department of Education.

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

## A APPENDIX

### A.1 VERBALIZATION LISTINGS

**Listing 2: Verbalization pattern of videos for the behavior understanding task:**

```
Input: <video> .. </video>
The video has the following scenes:
Scene 1: {ASR: Welcome to a quick tutorial , OCR: Adobe Premiere Pro, Captions: A desktop interface , Replays: 60},
Scene 2: {ASR: on using Premiere Pro to edit , Captions: A computer interface , with an image of a white horse. Objects − Horse,
        Grass, Fence., Replays: 53},
...
It was posted on Adobe's YouTube channel with the title 'Using Premiere Pro like a Pro' on Aug 15 2022. Adobe's YouTube
        channel has 100k subscribers . This video was viewed by 346 thousand people and liked (as a percentage of likes /views) by
        2.3% people. Why is the scene 23 one of the most replayed scenes?

Output: The scene shows the transformation of the image after the changes.
```

**Listing 3: Verbalization pattern of emails for the behavior domain adapation task. The email content and CTR is for demonstration purposes only.**

```
Input: Email with Subject: Lock it down before you send it out.
Header: Nobody messes with your PDFs.
Body text: Add password protection , secure encryption , and restricted editing to your PDFs with Adobe Acrobat Pro DC. Share
        only what you want and nothing more. A button that says 'Get started '. An image of a laptop , with window open on it .
        Image text : "Protect using password".
Foreground colors : grey , blue . Background colors: lavender , white . Image Emotions: security , serious . Image keywords: laptop ,
        protect , password, lock. Aesthetic value: low. Clutter level : medium. The email is created by a Creative Professional ,
        for the product Adobe Acrobat Pro. It is sent to users in the United States , in the commercial market. Specifically , it
        is sent to Power users with the intent of Active Use.
The email was sent 109 times between 25 August, 2022 and 26 August, 2022, and had a click through rate of [MASK]%.

Output: 0.037%.
```

**Listing 4: Verbalization pattern to teach behavior in the reverse direction (predicting content given behavior):**

```
Input: <video> .. </video> The video has the following scenes: Scene 1: {ASR: [MASK], Replays: 60%}, Scene 2: {ASR: with
        Premiere, Captions: Woman looking at screen , Replays: 34%},
...
Scene 5: {ASR: has never been, Captions: Colour Pallete , Replays: 47%},
Scene 6: {ASR: been easier , Captions: Colour Pallete , Replays: 54%},
...
It was posted on Adobe's YouTube channel with the title 'Using Premiere Pro like a Pro' on Aug 15 2022. It is viewed 203k
        times and liked 1.2%. Adobe's YouTube channel has 100k subscribers . Predict the masked ASR value for the masked scenes.

Output: Scene 1:{ASR: Welcome to a quick tutorial .}
```

**Listing 5: Verbalization pattern of videos for the content simulation task:**

```
Input: <video> .. </video> The video has the following scenes: Scene 1: {ASR: [MASK], Replays: 60%}, Scene 2: {ASR: with
        Premiere, Captions: Woman looking at screen , Replays: 34%},
...
Scene 5: {ASR: has never been, Captions: Colour Pallete , Replays: 47%},
Scene 6: {ASR: been easier , Captions: Colour Pallete , Replays: 54%},
...
It was posted on Adobe's YouTube channel with the title 'Using Premiere Pro like a Pro' on Aug 15 2022. It is viewed 203k
        times and liked 1.2%. Adobe's YouTube channel has 100k subscribers . Predict the masked ASR value for scene 1. Choose
        from the given options .
Option−1: Welcome to a quick tutorial ,
Option−2: Samsung Galaxy A20 smartphone,
...
Option−25: regulations . We haven't had.
```

**Listing 6: Verbalization pattern of videos for the behavior simulation task:**

```
Input: <video> .. </video> The video has the following scenes:
Scene 1: {ASR: Welcome to a quick tutorial , OCR: Adobe Premiere Pro, Captions: A desktop interface , Replays: [MASK]},
Scene 2: {ASR: on using Premiere Pro to edit , Captions: A computer interface , with an image of a white horse. Objects − Horse,
        Grass, Fence., Replays: [MASK] }, ...
It was posted on Adobe's YouTube channel with the title 'Using Premiere Pro like a Pro' on Aug 15 2022. Adobe's YouTube
        channel has 100k subscribers . Can you tell the replay values for scenes 2 to 5. How many times will this video be
        viewed and liked as a percentage of likes /views?

Output: Scene 1: {Replay: 60%}, Scene 2: {Replay: 85%}, ..., Views: 2.1 Million , Likes−per−View: 2.5%
```

**Listing 7: Verbalization pattern of Twitter posts for the behavior simulation task:**

```
Input: Given a tweet of pfizer posted by the account PfizerMed on 2023−01−12. Tweet : Announcing a new ASGCT−Pfizer grant to
        support independent medical education initiatives on genetic medicines. For details , click Request for Proposals . <
```

```
hyperlink>. Apply by January 30, 2022 # raredisease  #ASGCT #GeneTherapy <hyperlink>. Verbalisation of media content :  \"
caption\": \"A close−up of a DNA double helix, showcasing its  structure  and blue  color\",\"keywords\": \"DNA, double
helix, structure , blue, close−up, molecular biology , genetics , biology , scientific    illustration \"}. Predict whether it
will  recieve  high or low  likes ?",

Output: This tweet has low  likes .
```

Listing 8: Verbalization pattern of Twitter posts for the content simulation task:

```
Input :  Generate a tweet given the media verbalization and the likes it got. Tweet is for  pfizer  to be posted by the account
PfizerMed on 2023−01−12. Verbalisation  of media content : \"caption\": \"A close−up of  a DNA double helix, showcasing
its  structure  and blue  color\",\"keywords\": \"DNA, double helix,  structure , blue, close−up, molecular biology ,
genetics , biology ,  scientific    illustration \"}. This tweet has low  likes ."

Output: "Tweet :  Announcing a new ASGCT−Pfizer grant to support  independent medical education   initiatives  on genetic medicines
. For  details , click Request for Proposals . <hyperlink>. Apply by January 30, 2022 # raredisease  #ASGCT #GeneTherapy
<hyperlink>"}
```

## A.2  OTHER RELATED WORK

**Models of Human Communication:** Communication is the situation in which a source transmits a message to a receiver with conscious intent to affect the latter's behaviors (Osgood et al., 1957; Miller, 1966). Thus, in the most general terms, communication implies a sender, a channel, a message, a receiver, a relationship between sender and receiver, an effect, a context in which communication occurs and a range of things to which 'messages' refer (McQuail & Windahl, 2015; Lasswell, 1948). As per this, all of the content produced by humanity is essentially communication from a sender to a receiver over some channel and with some effect. Despite much research on communication in social sciences since the 1900s, there has been little adoption of it in machine learning modeling. A prime artefact of this is that the biggest models in machine learning (LLMs) are trained only on content (messages) and ignore other factors in communication (the intended receiver, channel, and behavior) even when they are available.

**Prior Efforts To Model Behavior:** While there has been much research in ML to model human behavior, it has been disconnected from language and, sometimes, real-world data. For instance, Agent-based modeling (ABMs), a popular paradigm in Reinforcement Learning, has been employed to model behavior (Bankes, 2002; Romero et al., 2023; Park et al., 2023). Nevertheless, ABMs tend to view humans as rational economic agents who communicate primarily through their actions, neglecting the significance of content in communication. In ABMs, agents strive to maximize their rewards, whereas communication does not always aim to optimize specific, well-defined reward signals. Moreover, the scarcity of large repositories containing extensive records of human actions poses a challenge when training ABMs to learn human behavior. Consequently, existing large models trained on human behavior, such as the ABMs and decision transformer and its variants, often rely on simulated data, such as game environments, rather than real human behavior (Chen et al., 2021). This reliance on artificially generated data introduces biases inherent to the creators of the training data, making it difficult to capture authentic human behavior. However, recent advancements have demonstrated the potential of large models trained on real-world tokens encompassing various modalities, like images, videos, audio, and text, as the basis for diverse tasks (Ge et al., 2023; Li et al., 2023a). Notably, LLMs, as exemplars of foundation models, have exhibited impressive performance across a range of tasks, including those they were not explicitly trained for, such as emotion recognition, named entity recognition, and complex tasks like table understanding (Ye et al., 2023; Bhattacharyya et al., 2023).

Further, there has also been much work in modeling behavior using conventional modeling techniques, such as regression, bagging and boosting (Mazloom et al., 2016; Villarroel Ordenes et al., 2019), neural networks (Ding et al., 2019; Wang et al., 2018; Khosla et al., 2014), and transformers (Wu & Krahenbuhl, 2021; Xiao et al., 2022). While these models can certainly model behavior, LLMs show generalization powers which extend to capabilities much beyond just behavior simulation. For instance, once trained on behavior tokens, other than behavior simulation, LLMs can now generate behavior optimized content (Table 2), explain behavior (Table 3), and domain-adapt to other behaviors (Table 6), none of which are shown by other models. The other concurrent works which model behavior using LLMs (Kang et al., 2023) model just behavior (for example, by CTR prediction) by attaching classification or regression heads to LLMs and thereby lose out on the text-to-text paradigm where LLMs show their best performance and generalization capabilities. In

addition, similar to non LLM paradigm, this method loses out on other capabilities like generating behavior optimized content and explaining behavior.

## A.3 REMAINING TABLES AND FIGURES

| Model | #Params | Training type | Training | RMSE | $R^2$ | Accuracy |
|-------|---------|---------------|----------|------|-------|----------|
| LCBM | | BFT | Replay values 3-masked | 1.31 | 0.87 | 15.89 |
| LCBM | 13B | BFT | Replay values 5-masked | 1.48 | 0.82 | 19.93 |
| LCBM | | BFT | Replay values 7-masked | 1.71 | 0.78 | 15.20 |
| LCBM | | BFT | Replay values 11-masked | 1.55 | 0.82 | 13.94 |
| GPT-4 | >100B[†] | ICL | 10-shot | 3.50 | -0.01 | 7.84 |
| GPT-4 | | ICL | 2-shot | 3.58 | -0.03 | 5.39 |
| GPT-3.5 | 175B | ICL | 3-shot | 64.40 | -256.96 | 2.48 |
| GPT-3.5 | | ICL | 2-shot | 64.88 | -375.83 | 1.27 |
| Random | - | - | - | 4.67 | 0 | 3.94 |

Table 4: **Behavior Simulation.** RMSE, $R^2$, and accuracy scores for like/view ratio prediction task. To calculate accuracy, the model is said to classify correctly if the absolute error between the predicted and ground truth likes/views is less than or equal to 10%. BFT denotes behavior fine-tuning, and ICL stands for in-context learning. Replay values $k$-masked means a model which is trained by masking $k$ consecutive values of the replay graph while doing BFT. We note that LCBM while being at least 10x smaller than the other models, performs the best. The best results over four runs are reported for all models. Best models are denoted in green and runner-ups in blue .

| Training | Model | #Params | Topic | Emotion | | Persuasion | Action | Reason |
|----------|-------|---------|-------|---------|---------|------------|--------|--------|
| | | | All labels | Clubbed | | | | |
| **Random** | **Random** | - | 2.63 | 3.37 | 14.3 | 8.37 | 3.34 | 3.34 |
| **Zero-shot** | **GPT-3.5** | 175B | 51.6 | 11.68 | 79.69 | 35.02 | 66.27 | 59.59 |
| | **Vicuna** | 13B | 11.75 | 10.5 | 68.13 | 26.59 | 20.72 | 21.00 |
| | **VideoChat** (Li et al., 2023b) | 13B | 9.07 | 3.09 | 5.1 | 10.28 | - | - |
| | **LCBM** | 13B | 42.17 | 7.08 | 58.83 | 32.83 | 39.55 | 27.91 |

Table 5: **Content Understanding.** Comparison of several models, including behavior instruction tuned models before and after BFT. We compare the models across topic, emotion, and persuasion strategy detection tasks as per the framework given by Bhattacharyya et al. (2023). We see that our model outperforms similarly sized models (Vicuna, VideoChat) in most tasks. Best models are denoted in green and runner-ups in blue .

---

[†]The exact size of GPT-4 is unknown.

[‡]Brand Separated means that the train and test set don't have any overlap in terms of brands, Time Separated means that the test set starts after the last tweet in the train set. BFT denotes behavior fine-tuning, and ICL stands for in-context learning. The best results over four runs are reported for all models. Best models are denoted in green and runner-ups in blue .

| In-house Email Marketing | | | | | | | |
|---|---|---|---|---|---|---|---|
| **LCBM Type** | **Fine-tuned** | **Trained On** | | | **Tested On** | **RMSE** | **$R^2$** |
| | **on YouTube?** | **Unique Emails** | **Unique Segments** | **Email-Segment Pairs** | | | |
| Domain-Adapted In-Domain | Yes | 100 | 10 | 1k | Different Segment (emails could be same) | 14.47 | 0.64 |
| | No | 600 | 560k | 350k | | 25.28 | 0.55 |
| Domain-Adapted In-Domain | Yes | 100 | 10 | 1k | Different Segments & Different Emails | 27.28 | 0.54 |
| | No | 600 | 560k | 350k | | 29.28 | 0.5 |

| LVU Benchmark | | | |
|---|---|---|---|
| **Training** | **Model** | **Testing** | **MSE** |
| Trained | R101-slowfast+NL (Wu & Krahenbuhl, 2021) | Test set | 0.386 |
| Trained | VideoBERT (Sun et al., 2019) | Test set | 0.32 |
| Trained | Qian et al. (2021) | Test set | 0.353 |
| Trained | Xiao et al. (2022) | Test set | 0.444 |
| Trained | Object Transformers (Wu & Krahenbuhl, 2021) | Test set | 0.23 |
| Zero-shot | LCBM (Ours) | Test set | 0.14 |
| Zero-shot | GPT-3.5 | Test set | 0.03 |
| Zero-shot | Vicuna | Complete dataset | 0.44 |
| Zero-shot | LCBM (Ours) | Complete dataset | 0.30 |
| Zero-shot | GPT-3.5 | Complete dataset | 0.02 |

Table 6: **Behavior Domain Adaptation.** We test the generalization capability of LCBM on two tasks: (1) Behavior simulation on In-house Email Marketing Data, (2) Behavior simulation on the LVU benchmark. For (1), we train two versions of LCBM with the In-house Email Marketing data: one was trained on YouTube videos and further BFT on a few email samples (*domain-adapted*), and the other was BFT on a larger set of emails, but not including YouTube data (*in-domain*)[§]. We report the RMSE and $R^2$ scores for this task. For (2), we compare LCBM with other state-of-the-art results and GPT-3. In (1), we note that the domain-adapted LCBM performs better than the in-domain LCBM in both settings. We posit that YouTube data helps LCBM understand how a company's viewers like to hear from it, giving LCBM an edge over a model trained on a small amount of the same data (600 unique emails). In (2), LCBM performs better than the existing state-of-the-art. Surprisingly, GPT-3.5 does better than LCBM on this task. From both (1) and (2), we gather that a model trained on certain YouTube behaviors performs better on other behaviors, thus showing promise of domain-adaptation in the behavior modality. Best models are denoted in green and runner-ups in blue .

| Model | #Params | Training type | Training | Time Separated | Brand Separated |
|---|---|---|---|---|---|
| GPT-3.5 | 175B | ICL | Few-shot | 58.84 | 64.19 |
| LCBM | 13B | BFT | Twitter | 74.3 | 97.69 |
| LCBM | 13B | BFT | Twitter and YouTube data | 76.87 | 92.19 |

Table 7: **Behavior Simulation and Behavior Domain Adaptation**[‡]. Two-way classification accuracies for like prediction on Twitter. Given content, channel, and time, predict behavior (High, Low). We note that LCBM trained on Twitter and YouTube performs better than the one trained only on Twitter, showing signs of performance improvement by domain adaptation.

| Model | Training | Test | BLEU-1 | BLEU-2 | BLEU-3 | BLEU-4 | ROUGE-l |
|---|---|---|---|---|---|---|---|
| GPT-3.5 | ICL | Brand Separated | 53.95 | 42.36 | 31.84 | 24.28 | 15.24 |
| | | Time Separated | 57.69 | 45.11 | 33.67 | 25.52 | 15.27 |
| LCBM | BFT on Twitter | Brand Separated | 62.29 | 46.59 | 33.98 | 25.64 | 14.44 |
| | | Time Separated | 70 | 54.4 | 41.43 | 32.48 | 17.38 |
| LCBM | BFT on Twitter + Youtube | Brand Separated | 64.28 | 48.1 | 35.17 | 26.63 | 14.83 |
| | | Time Separated | 70.23 | 54.54 | 41.52 | 32.54 | 17.45 |

Table 8: **Content Simulation and Behavior Domain Adaptation**[‡]. Given behavior, channel, time, tweet media caption as prompt, predict content (tweet text). We note that LCBM trained on Twitter and YouTube performs better than the one trained only on Twitter, showing signs of performance improvement by domain adaptation.

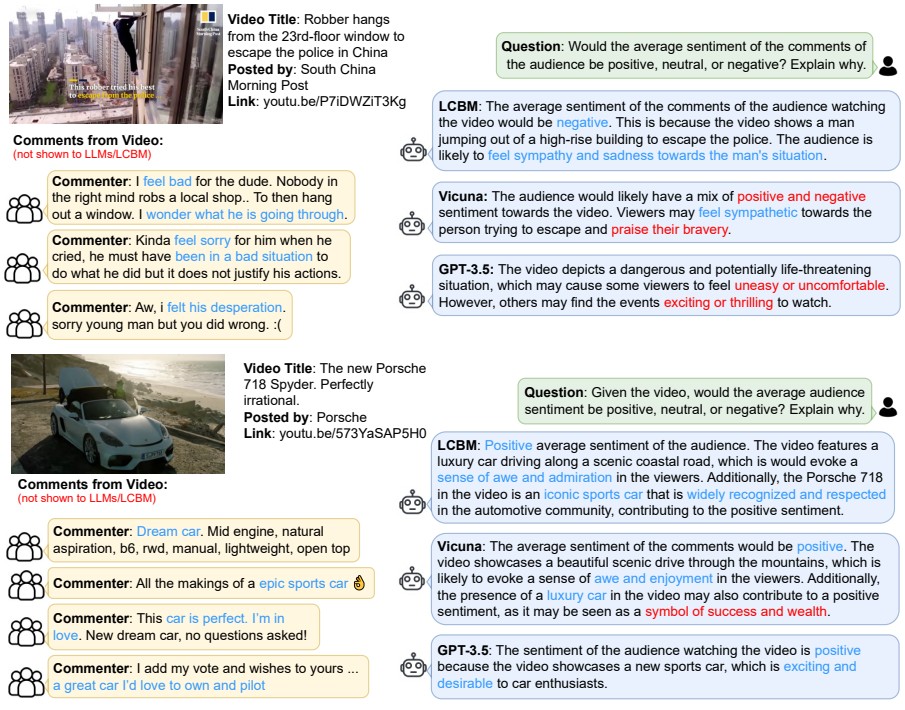

Figure 5: A few examples showing LCBM's ability to understand and explain human behavior of audience sentiment. We also compare it against other models like Vicuna and GPT-3.5.

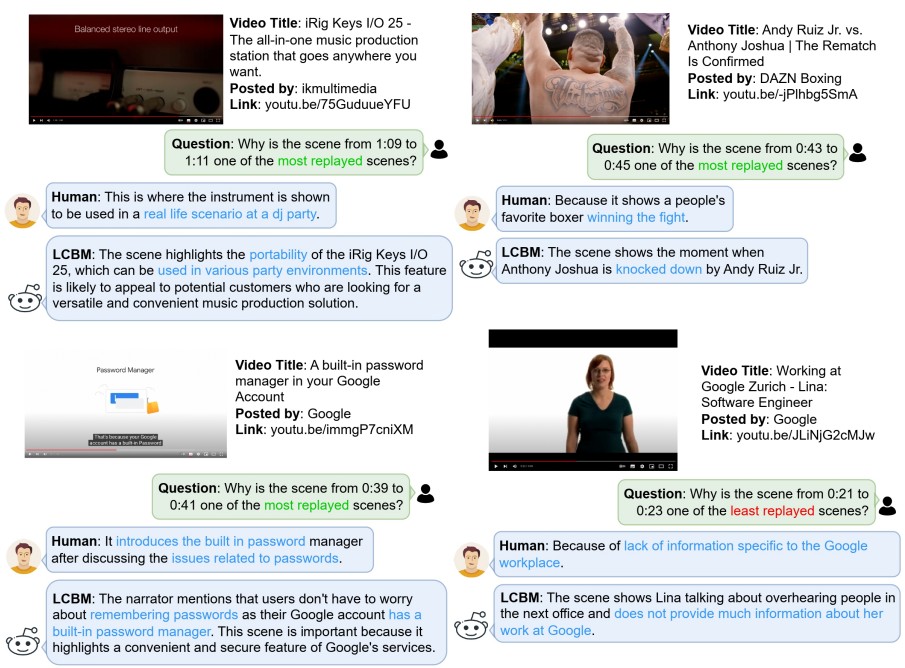

Figure 6: A few examples showing LCBM's ability to understand and explain human behavior of scene replayability. We compare it against human-provided explanations of the same.

| | |
|---|---|
| **Date Range** | April 1, 2022 to June 12, 2023 |
| **Number of Countries** | 225 |
| **Target Products** | Top Products used by millions of users |
| **Customers Segmented on the basis of** | Type of use, user expertise, frequency of use, and others |

Table 9: Details of the In-house Marketing Email dataset used to evaluate behavior generalization capabilities of the LCBM

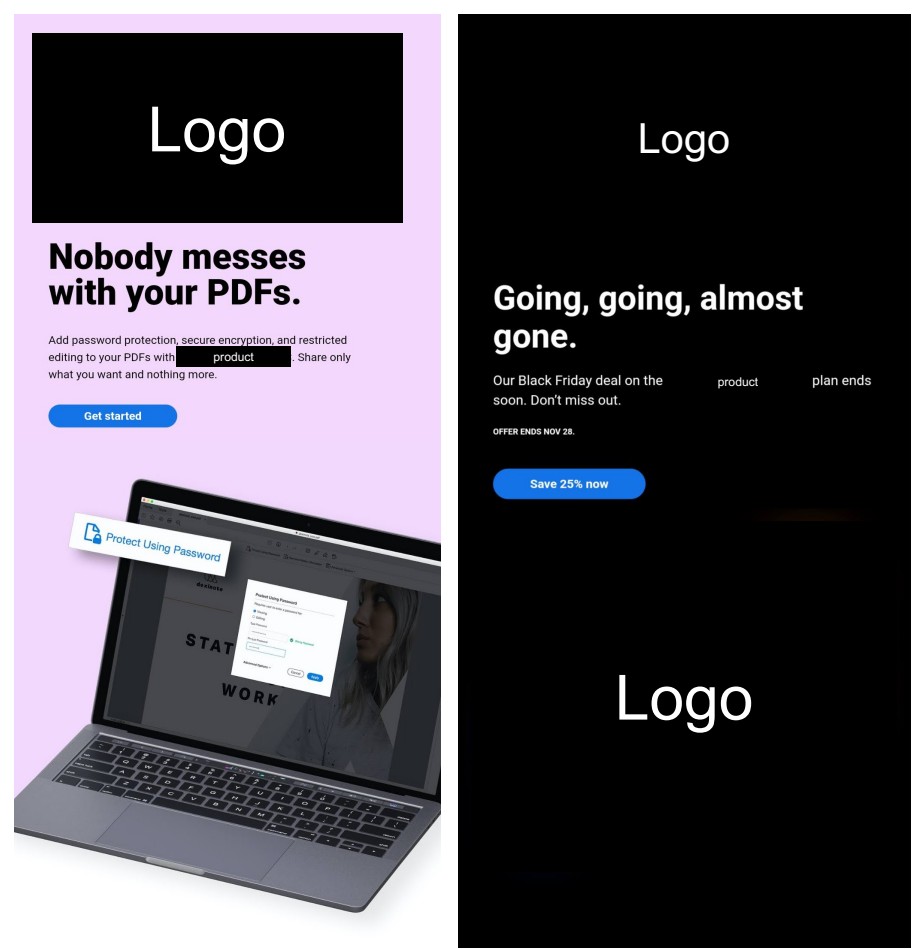

Figure 7: The In-house marketing emails used in the Email dataset look similar to the ones shown here.

