# OpenReview forum: "Large Content And Behavior Models To Understand, Simulate, And Optimize Content And Behavior"
_ICLR.cc/2024/Conference — ICLR 2024 spotlight_

### Official Review · Reviewer_aTeR · 2023-10-27

**Soundness:** 4 excellent
**Presentation:** 4 excellent
**Contribution:** 4 excellent
**Rating:** 10
**Confidence:** 3

**Summary:**

In this paper, the authors attempts to tackle the challenge of communication effectiveness by LLMs. The authors posed the effectiveness problem as a text-to-text tokens by including "receiver behavior" as a new modality and introduce "behavior tokens" to LLMs. The authors made several contributions towards tackling this problem (1) the authors created a large corpus of data (CBC) containing Youtube video segments as well as viewer retention/replay values and number of viewers and likes, as well as communicator information (channel name and subscriber counts), and (2) a comprehensive test benchmark for content behavior, including behavior prediction, content prediction, behavior understanding, content understanding, and behavior domain adaptation to non-CBC domains (such as email marketing), and (3) The authors introduced a new method called Behavior Fine-tuning, and used it to fine-tune a 13B parameter model (LCBM), and the resulting model outperformed much larger models such as GPT-3.5 and GPT-4 on the content behavior test benchmark.

**Strengths:**

1. This paper made significant contribution to an important problem: teaching LLMs to understand and reason about communication effectiveness. The authors supported the significance of the problem from Shanon and Weaver's Information Theory, and they carefully posed this problem into a text-to-text problem so researchers can now explore ways to tackle this problem via LLMs.

2. The paper introduced a large dataset of behavior-content data, as well as a test benchmark that comprehensively evaluates an LLM's ability to reason about content behavior and communication effects (including some human-annotated ground truths). These contributions will be very valuable for future researchers working on communication effectiveness or other related problems.

3. The paper introduced a novel method to fine-tune LLMs for behavior understanding, and their resulting 13B parameter model outperformed GPT-4/GPT-3.5, often by large margins, on most tasks in content behavior test benchmark, which is really impressive.

4. The figures in the paper are really informative and can really help readers better understand the problem, LCBM, and tasks in the test set.

**Weaknesses:**

Overall this is a really nice paper. A very minor weakness of this paper may be that the font sizes of the figures and tables are really small and sometimes inconsistent between different tables and figures. I understand that this is likely due to space limit in the main paper, so perhaps moving some parts of the detailed discussions in the introduction (such as the backgrounds in "How do we solve the effectiveness problem while retaining the other two levels?") to the appendix would alleviate the problem.

**Questions:**

(1) Is BFT training using CBC data? Is the test benchmark also using CBC data? If both are using CBC data, is there a train/test split within CBC so a different split is used for training and testing?

(2) In section 3, the paper mentioned that the behavior tokens are not used for GPT-3.5 and GPT-4. What about the visual tokens about the videos? Did you find a way to input them to GPT, or did you passed in the visual content of the video through some different means?

---

> ### Author Response · Authors · 2023-11-16
>
> Thank you for reviewing our paper and for your insightful questions 🙂.
>
> Regarding the figure and table font sizes, we apologize for the inconsistency. We plan to address this by relocating some sections to the appendix and ensuring uniform font sizes throughout.
>
> Regarding your point (1) on Behavior Finetuning:
>
> We utilize the train subset of the CBC data for Behavior Finetuning. Here is the breakdown for different tasks using various subsets of the CBC data:
>
> 1. Behavior Simulation:
>
>     a. YouTube Videos: 2027 videos were selected without temporal overlap with the train set, encompassing diverse brands, industries, dates, and receiver behavior.
>
>     b. Twitter Tweets (new dataset):
>
>     * Brand Separated: 1557 tweets from diverse brands, industries, dates, and receiver behavior.
>
>     * Time Separated: 782 tweets without temporal overlap with the train set from diverse brands, industries, dates, and receiver behavior.
>
> 2. Content Simulation:
>
>     a. YouTube Videos: 2027 videos chosen without temporal overlap with the train set.
>
>     b. Twitter Tweets (new dataset):
>
>     * Brand Separated: 1557 tweets sampled from diverse brands, industries, dates, and receiver behavior.
>
>     * Time Separated: 782 tweets without temporal overlap with the train set from diverse brands, industries, dates, and receiver behavior.
>
> 3. Behavior Domain Adaptation Tasks:
>
>     a. Emails Corpus: 3000 email-segment pairs without temporal overlap between train and test sets, using emails until July 2022 for training and testing on emails from August 2022 onwards.
>
>     b. LVU Benchmark [1]: Zero-shot testing on the entire dataset without prior training.
>
> 4. Content Understanding Tasks:
>
>     a. Advertisement Understanding Corpus (Hussain et al, 2017): Zero-shot testing on the entire dataset without training.
>
>     b. Persuasion Strategy Corpus (Bhattacharyya et al, 2023): Zero-shot testing on the entire dataset without training.
>
> 5. Behavior Understanding Tasks:
>
>     a. Replay Behavior Understanding: No training, 200 videos randomly sampled from the behavior simulation test set.
>
>     b. Comment Sentiment Understanding: No training, 200 videos randomly sampled from the behavior simulation test set.
>
> Additionally, subsequent to paper submission, we've compiled a large-scale content behavior dataset comprising images and corresponding human behavior extracted from 168 million tweets across 10135 enterprise Twitter accounts from 2007 to 2023. This dataset includes account names, tweet text, associated media verbalizations (including image captions, keywords, colors, and tones), tweet timestamps, and like counts. We plan to release both the YouTube and Twitter corpora. Our experiments with behavior simulation, domain adaptation, and content simulation on this corpus yield results consistent with those reported in the paper:
>
> **Behavior Simulation. Give content, channel, time, predict behavior (High, Low). Reported values are 2-way classification accuracies**
>
> (Brand Separated means that the train and test set don't have any overlap in terms of brands, Time Separated means that the test set starts after the last tweet in the train set)
>
> | **Model** | **Behavior Training** | **Behavior Datasets Included In Training** | **Brand Separated** | **Time Separated** |
> |---|---|---|---|---|
> | GPT-3.5 (few-shot) | In-context-learning |  | 58.84 | 64.19 |
> | LCBM | Finetuning | Twitter (77.3k tweets) | 74.3 | **97.69** |
> | LCBM | Finetuning | Twitter (77.3k tweets) + Youtube (40k) data | **76.87** | 92.19 |
>
>
>
>
> **Content Simulation Experiment. Given behavior, channel, time, tweet media caption as prompt, predict content (tweet text)**
>
> | Model | Behavior Training | Behavior Datasets Included In Training | Test-split | Bleu-1 | Bleu-2 | Bleu-3 | Bleu-4 | Rouge-l |
> |---|---|---|---|---|---|---|---|---|
> | GPT-3.5 (few-shot) | In-context-learning |  | Brand Separated | 53.95 | 42.36 | 31.84 | 24.28 | 15.24 |
> |  | In-context-learning |  | Time Separated | 57.69 | 45.11 | 33.67 | 25.52 | 15.27 |
> | LCBM | Finetuning | Twitter (77.3k tweets) | Brand Separated | 62.29 | 46.59 | 33.98 | 25.64 | 14.44 |
> |  | Finetuning | Twitter (77.3k tweets) | Time Separated | 70 | 54.4 | 41.43 | 32.48 | 17.38 |
> | LCBM | Finetuning | Twitter (77.3k tweets) + Youtube (40k) data | Brand Separated | **64.28** | **48.1** | **35.17** | **26.63** | **14.83** |
> |  | Finetuning | Twitter (77.3k tweets) + Youtube (40k) data | Time Separated | **70.23** | **54.54** | **41.52** | **32.54** | **17.45** |
>
>
> (continued in the next comment)

---

> ### Author Response · Authors · 2023-11-16
>
> About inputting the visual tokens to GPT-3.5 and 4: we verbalize videos and images and feed them to GPT-3.5 and 4. We test LCBM in the same conditions and find that there is little difference in performance if visual tokens are not given.
>
> For instance, we show results on the Replay Value Prediction task of Behavior Simulation. Similar results are observed on other tasks. Thanks for pointing this out; we will add these results to the appendix in the next version.
>
> | **Model** | **Past** | **** | **Future** | **** | **Random - 5 missed** | **** | **Random - 7 missed** | **** |
> |---|---|---|---|---|---|---|---|---|
> |  | RMSE | Accuracy | RMSE | Accuracy | RMSE | Accuracy | RMSE | Accuracy |
> | LCBM (with video tokens) | 11.53 | 52.06 | 12.02 | 53.06 | 8.13 | 64.83 | 9.22 | 60.26 |
> | LCBM (without video tokens) | 12.21 | 49.77 | 11.08 | 52.51 | 7.66 | 65.42 | 9.44 | 58 |
> | GPT-4 | 35.05 | 19.34 | 18.07 | 39.33 | 17.42 | 38.1 | 21.26 | 28.05 |
> | GPT-3.5 | 34.1 | 19.06 | 24.71 | 27.14 | 24.52 | 24.81 | 26.3 | 18.74 |
>
>
> Additionally, we show results on the Likes-to-Views ratio task of Behavior Simulation.
>
> | **Model** | **RMSE** | **R2** | **Accuracy** |
> |---|---|---|---|
> | LCBM (with video tokens) | 1.31 | 0.87 | 15.89 |
> | LCBM (without video tokens) | 1.36 | 0.85 | 18.28 |
> | GPT-4 | 3.5 | -0.01 | 7.84 |
> | GPT-3.5 | 64.4 | -256.96 | 2.48 |
>
>
> Verbalization pattern of videos for the behavior understanding task, without using video tokens:
> ```
> The video has the following scenes:
> Scene 1: {ASR: Welcome to a quick tutorial , OCR: <product>, Captions: A desktop interface , Replays: 60},
> Scene 2: {ASR: on using <product> to edit , Captions: A computer interface , with an image of a white horse. Objects − Horse, Grass,
> Fence., Replays: 53}, ...
> It was posted by <channel> with the title ' Using <product> like a Pro ' on Aug 15 2022. <channel> has 100k subscribers . This video was viewed by 346 thousand people and liked (as a percentage of likes /views) by 2.3% people.
> ```
>
>
>
>
>
> [1] Chao-Yuan Wu, Philipp Krahenb, Towards Long-Form Video Understanding, CVPR 2021, Link: https://openaccess.thecvf.com/content/CVPR2021/papers/Wu_Towards_Long-Form_Video_Understanding_CVPR_2021_paper.pdf

---

### Official Review · Reviewer_A7CN · 2023-10-30

**Soundness:** 3 good
**Presentation:** 2 fair
**Contribution:** 2 fair
**Rating:** 5
**Confidence:** 3

**Summary:**

This paper studies the problem of behavior simulation, content simulation, and behavior domain adaptation in LLMs. The authors train LLMs on behavior tokens to show that other than content understanding tasks, the trained models are now able to have good performance across all the behavior-related tasks as well.

**Strengths:**

1. Formulating the problem as behavior/content prediction and understanding is interesting;

2. The authors systematically studied the behavior/content related problems in online website domains, and train an LLM to solve these problems;

3. The experiments demonstrates promising results for simulating and understanding tasks.

**Weaknesses:**

1. The authors claimed that the paper studies the problem of predicting and optimizing communication for desired receiver behaviors and intents, i.e., the effectiveness level of communication, but what the authors actually did in this paper is simply exploring the Youtube and email datasets. As far as I understand, the key to improve the performance of behavior/content prediction and understanding is to improve the general reasoning ability of LLMs, rather than finetuning LLMs on specific datasets. From this perspective, the contribution of this paper is very limited.

2. In terms of presentation, I think the authors make things too complicated by discussing the effectiveness and behavior issues, which make the readers hard to quickly get the main idea and contribution of this paper.

3. The behavior simulation task sounds like a data mining problem, e.g. predicting whether a user will watch a video. LLMs are not quite suitable for such tasks because they are too slow and they cannot make use of massive user-item interaction data.

**Questions:**

See weaknesses above.

---

> ### Author Response · Authors · 2023-11-16
>
> Thank you for your review and interesting questions 🙂.
>
> Regarding (1), although we acknowledge the potential of enhancing the general reasoning ability of Language Models (LLMs) to address the issue of communication effectiveness, our study presents compelling evidence that directly exposing LLMs to behavior tokens alongside content tokens can be a more effective method to instill behavioral understanding. Our approach and underlying rationale for fine-tuning diverge significantly from established methodologies. Specifically:
>
> 1. While GPT-3.5 and GPT-4 have demonstrated superior general comprehension across various benchmarks [1,2,3], our research illustrates that training an LLM on behavior tokens enables a 13B model to surpass much larger models—despite their stronger general understanding—across all tasks. This performance extends to both tasks specifically fine-tuned and those that were not originally targeted for fine-tuning.
>
> 2. In-context learning, although impressively successful in other domains, proves ineffective in the behavior domain. Large in-context-finetuned language models (GPT-3.5, GPT-4, etc.) exhibit performance comparable to a random baseline, indicating their lack of exposure to behavior tokens.
>
> 3. Incorporating behavior tokens in content and behavior simulations not only enhances behavior and content simulation (as anticipated) but also unexpectedly improves behavior domain-adaptation (Table-6), behavior-understanding (Table 4, Figs. 7, 8), and content understanding (Table-5). Notably, LCBM exhibits enhancements in these tasks despite not being explicitly trained for them. This aligns with how LLMs, trained for next word prediction, showcase proficiency in various tasks such as question answering, summarization, and natural language inference. We demonstrate these capabilities across three behavior domain datasets (YouTube, Emails, and LVU) and five tasks spanning two content domain datasets.
>
> 4. Presently, large language models focus on training with content tokens (text, vision, videos, etc.) while disregarding behavior tokens (Biderman et al 2023, Penedo et al 2023). Consequently, LLMs lack behavior simulation (Table 1, 2), content simulation (generation) (Table 3), and behavior understanding (Table 4) capabilities.
>
> Further, we show our results on four datasets beyond Youtube:
> 1. Long Video Understanding (LVU) Benchmark
> 2. Emails dataset
> 3. Two Advertisement Understanding Datasets
> 4. Additionally, subsequent to paper submission, we've compiled a large-scale content behavior dataset comprising images and corresponding human behavior extracted from 168 million tweets across 10135 enterprise Twitter accounts from 2007 to 2023. This dataset includes account names, tweet text, associated media verbalizations (including image captions, keywords, colors, and tones), tweet timestamps, and like counts. We plan to release both the YouTube and Twitter corpora. Our experiments with behavior simulation, domain adaptation, and content simulation on this corpus yield results consistent with those reported in the paper:
>
>
> **Behavior Simulation. Give content, channel, time, predict behavior (High, Low). Reported values are 2-way classification accuracies**
> (Brand Separated means that the train and test set don't have any overlap in terms of brands, Time Separated means that the test set starts after the last tweet in the train set)
>
> | **Model** | **Behavior Training** | **Behavior Datasets Included In Training** | **Brand Separated** | **Time Separated** |
> |---|---|---|---|---|
> | GPT-3.5 (few-shot) | In-context-learning |  | 58.84 | 64.19 |
> | LCBM | Finetuning | Twitter (77.3k tweets) | 74.3 | **97.69** |
> | LCBM | Finetuning | Twitter (77.3k tweets) + Youtube (40k) data | **76.87** | 92.19 |
>
>
>
>
> **Content Simulation Experiment. Given behavior, channel, time, tweet media caption as prompt, predict content (tweet text)**
>
> | Model | Behavior Training | Behavior Datasets Included In Training | Test-split | Bleu-1 | Bleu-2 | Bleu-3 | Bleu-4 | Rouge-l |
> |---|---|---|---|---|---|---|---|---|
> | GPT-3.5 (few-shot) | In-context-learning |  | Brand Separated | 53.95 | 42.36 | 31.84 | 24.28 | 15.24 |
> |  | In-context-learning |  | Time Separated | 57.69 | 45.11 | 33.67 | 25.52 | 15.27 |
> | LCBM | Finetuning | Twitter (77.3k tweets) | Brand Separated | 62.29 | 46.59 | 33.98 | 25.64 | 14.44 |
> |  | Finetuning | Twitter (77.3k tweets) | Time Separated | 70 | 54.4 | 41.43 | 32.48 | 17.38 |
> | LCBM | Finetuning | Twitter (77.3k tweets) + Youtube (40k) data | Brand Separated | **64.28** | **48.1** | **35.17** | **26.63** | **14.83** |
> |  | Finetuning | Twitter (77.3k tweets) + Youtube (40k) data | Time Separated | **70.23** | **54.54** | **41.52** | **32.54** | **17.45** |
>
>
> (Continued in the next comment)

---

> ### Author Response · Authors · 2023-11-16
>
> Regarding (2) (presentation), we regret any confusion caused. Could you kindly specify the specific figures or sections that posed challenges for clarity in the upcoming version?
>
> Re (3), while LLMs might be slow for certain specific tasks, we believe they present a more general framework, which has empirically shown much better content understanding and generation capabilities than any other systems historically. This motivates us to use LLMs (as compared to other competitive technologies) for behavior understanding (the next step and the third level of communication).
>
> - To enable LLMs to learn behavior and content, following the T5 methodology, we convert behavior learning task to a “text-to-text” problem. We show that a model trained on ~2 trillion tokens (Llama), can be taught behavior with a few billion additional tokens. This shows that an adequately trained LLM might be taught a behavior as a new modality relatively easily.
>
> - A similar evidence is also provided by LLMs trying to learn the vision modality, such as Llava, VideoLlama, Minigpt-4, and others [4,5,6], where they tune an LLM for vision modality with a few billion vision tokens (similar scale as us).
>
>
>
> [1] MMLU Benchmark: https://paperswithcode.com/sota/multi-task-language-understanding-on-mmlu
>
> [2] Winogrande Benchmark: https://paperswithcode.com/sota/common-sense-reasoning-on-winogrande
>
> [3] HellaSwag Benchmark: https://paperswithcode.com/sota/sentence-completion-on-hellaswag
>
> [4] Haotian Liu, Chunyuan Li, Qingyang Wu, and Yong Jae Lee. Visual instruction tuning
>
> [5] Yuying Ge, Yixiao Ge, Ziyun Zeng, Xintao Wang, and Ying Shan. Planting a seed of vision in large language model
>
> [6] Minigpt-4: Enhancing vision-language understanding with advanced large language models.

---

### Official Review · Reviewer_jTBK · 2023-10-31

**Soundness:** 4 excellent
**Presentation:** 3 good
**Contribution:** 4 excellent
**Rating:** 8
**Confidence:** 3

**Summary:**

The paper presents a novel and interesting question: predicting and optimizing communication to get the desired receiver behavior for intent alignment. The paper uses behavior tokens in the training corpora to guide models to finish behavior simulation, content simulation, and behavior domain adaptation tasks. Based on existing multi-modal LLMs, the proposed methods achieve promising results on several benchmarks such as LVU and Email Marketing Data.

**Strengths:**

Using behavior tokens to guide the model to finish various tasks is novel and important to me. In fact, "human:, assistant" question-answer pairs used in modern SFT models is also a kind of behavior token. The paper extends the scope to a broader concept. The experimental results are convincing. I believe that this paper has the potential to provide valuable insights to the field of intention alignment in large-scale language models, extending beyond the scope of video click-through rate as described in the paper.

**Weaknesses:**

Similar to Instruction Following Finetuning (SFT), it demonstrates that a small amount of data can activate the model's capabilities, resulting in more human-like responses. It would be interesting to study the impact of varying data quantities on the results in the context of this work, particularly in terms of CBC (Contextual Bandit Control). Additionally, exploring the influence of data quantity on the results when behavior tokens are present would also be insightful. Investigating these aspects would provide a further understanding of the relationship between data volume and the performance outcomes in this study.

**Questions:**

See weakness.

---

> ### Author Response · Authors · 2023-11-17
>
> Thank you for your thoughtful review and interesting insights 🙂.
> We agree that alignment mechanisms, such as through SFT and RLHF on question-answer pairs, is a kind of behavior / intent alignment. We will add this to the paper. It would be an interesting lens for the related work.
>
> In the limited rebuttal time, we tried varying the amount of data on some of the results, We will show more complete set of results in the next version of the paper.For instance, we show results on the Replay Value Prediction task of Behavior Simulation.
>
> We vary our dataset size with 5k and 10k unique videos, and compare performance on the behavior simulation task. We find that increasing dataset size helps in achieving better performance.
>
> | **Model** | **Past** | |**Future**| | **Random 5** | | **Random 7**  | |
> |---|---|---|---|---|---|---|---|---|
> | **** | RMSE | Accuracy | RMSE | Accuracy | RMSE | Accuracy | RMSE | Accuracy |
> | **LCBM (5k videos)** | 29.37 | 13.93 | 14.37 | 40.88 | 11.4 | 47.62 | 13.87 | 41.26 |
> | **LCBM (10k videos)** | 11.53 | 52.06 | 12.02 | 53.06 | 8.13 | 64.83 | 9.22 | 60.26 |
> | **** |  |  |  |  |  |  |  |  |
> | **GPT-4** | 35.05 | 19.34 | 18.07 | 39.33 | 17.42 | 38.1 | 21.26 | 28.05 |
> | **GPT-3.5** | 34.1 | 19.06 | 24.71 | 27.14 | 24.52 | 24.81 | 26.3 | 18.74 |
>
> Additionally, we show results on the Likes-to-Views ratio prediction task of Behavior Simulation.
>
>
> | **Model** | **RMSE** | **R2** | **Accuracy** |
> |---|---|---|---|
> | **LCBM (5k videos)** | 1.69 | 0.79 | 12.45 |
> | **LCBM (10k videos)** | 1.31 | 0.87 | 15.89 |
> | **** |  |  |  |
> | **GPT-4** | 3.5 | -0.01 | 7.84 |
> | **GPT-3.5** | 64.4 | -256.96 | 2.48 |
>
>
> Thanks for pointing this out, we will add more such results to the appendix in the next version.

---

> > ### Comment · Reviewer_jTBK · 2023-11-18
> > **Respone to Authors**
> >
> > I have read the response and maintain the same score.

---

> > ### Comment · Reviewer_jTBK · 2023-11-18
> > **Respone to Authors**
> >
> > I have read the response and maintain the same score.

---

> > > ### Author Response · Authors · 2023-11-18
> > >
> > > Thank you for taking out time and responding to our review!

---

### Official Review · Reviewer_73Nf · 2023-11-01

**Soundness:** 3 good
**Presentation:** 3 good
**Contribution:** 3 good
**Rating:** 6
**Confidence:** 3

**Summary:**

This paper aims to explore the content-behavior understanding abilities of LLMs in modern communication systems and proposes large content behavior models(LCBMs) to this end. Motivated by the recent vision-language model, LCBMs formulate the YouTube data (such as video frames, channel name, shares and likes) as the multimodal input sequence, and finetune a LLM in an autoregressive manner. Also, the authors release their generated data and a benchmark to spur the research purpose. Extensive results on five tasks show better performance of the proposed model than other baselines, including GPT-3.5 and GPT-4.

**Strengths:**

1) The task that employs the LLM to perform the content-behavior understanding sounds interesting. This provides new application directions to study the decision process of the LLMs.

2) The released content-behavior datasets and benchmarks contribute to a better community, which facilitates the follow-up research.

3)  Extensive experiments are conducted to test the performance of the proposed model, which shows the generalization capabilities on content-behavior understanding.

**Weaknesses:**

1) One of the main concerns is the technique novelty. From my understanding, the proposed model is mostly based on InstructBLIP[1] and only the GMHRA is newly introduced to aggregate the time dimension in video frames.

2) More experiment details need to be described. Is the baseline models are finetuned using the behavior instruction datasets ? If not, then it may be unfair to compare the proposed model with these baseline models.



[1] Dai et.al. InstructBLIP: Towards General-purpose Vision-Language Models with Instruction Tuning.

**Questions:**

1) Please provide more detailed information on how the visual encoder handles video input.

---

> ### Author Response · Authors · 2023-11-16
>
> Thank you for your insightful review! 🙂
>
> About (1) technique novelty, we agree with your review. Other than showing how to do behavior finetuning (using content and behavior instruction tokens), we do not intend to go for any other technique novelty in this paper because we find existing techniques are good enough to train our model well. Rather, we focus on the following contributions:
>
> 1. Incorporating behavior tokens in content and behavior simulations not only enhances behavior and content simulation (as anticipated) but also unexpectedly improves behavior domain-adaptation (Table-6), behavior-understanding (Table 4, Figs. 7, 8), and content understanding (Table-5). Notably, LCBM exhibits enhancements in these tasks despite not being explicitly trained for them. This aligns with how LLMs, trained for next word prediction, showcase proficiency in various tasks such as question answering, summarization, and natural language inference. We demonstrate these capabilities across three behavior domain datasets (YouTube, Emails, and LVU) and five tasks spanning two content domain datasets.
>
> 2. Presently, large language models focus on training with content tokens (text, vision, videos, etc.) while disregarding behavior tokens (Biderman et al 2023, Penedo et al 2023). Consequently, LLMs lack behavior simulation (Table 1, 2), content simulation (generation) (Table 3), and behavior understanding (Table 4) capabilities.
>
> 3. In-context learning, although successful in other domains, proves ineffective in the behavior domain. Large in-context-finetuned language models (GPT-3.5, GPT-4, etc.) exhibit performance comparable to a random baseline, indicating their lack of exposure to behavior tokens.
>
> 4. Next, to spur research on the topic of content and behavior models, we release behavior and content tokens collected over 40,000 public domain YouTube videos containing a variety of behavior tokens.
>
> 5. Additionally, subsequent to paper submission, we've compiled a large-scale content behavior dataset comprising images and corresponding human behavior extracted from 168 million tweets across 10135 enterprise Twitter accounts from 2007 to 2023. This dataset includes account names, tweet text, associated media verbalizations (including image captions, keywords, colors, and tones), tweet timestamps, and like counts. We plan to release both the YouTube and Twitter corpora. Our experiments with behavior simulation, domain adaptation, and content simulation on this corpus yield results consistent with those reported in the paper:
>
> **Behavior Simulation. Give content, channel, time, predict behavior (High, Low). Reported values are 2-way classification accuracies**
>
> (Brand Separated means that the train and test set don't have any overlap in terms of brands, Time Separated means that the test set starts after the last tweet in the train set)
>
> | **Model** | **Behavior Training** | **Behavior Datasets Included In Training** | **Brand Separated** | **Time Separated** |
> |---|---|---|---|---|
> | GPT-3.5 (few-shot) | In-context-learning |  | 58.84 | 64.19 |
> | LCBM | Finetuning | Twitter (77.3k tweets) | 74.3 | **97.69** |
> | LCBM | Finetuning | Twitter (77.3k tweets) + Youtube (40k) data | **76.87** | 92.19 |
>
>
>
>
> **Content Simulation Experiment. Given behavior, channel, time, tweet media caption as prompt, predict content (tweet text)**
>
> | Model | Behavior Training | Behavior Datasets Included In Training | Test-split | Bleu-1 | Bleu-2 | Bleu-3 | Bleu-4 | Rouge-l |
> |---|---|---|---|---|---|---|---|---|
> | GPT-3.5 (few-shot) | In-context-learning |  | Brand Separated | 53.95 | 42.36 | 31.84 | 24.28 | 15.24 |
> |  | In-context-learning |  | Time Separated | 57.69 | 45.11 | 33.67 | 25.52 | 15.27 |
> | LCBM | Finetuning | Twitter (77.3k tweets) | Brand Separated | 62.29 | 46.59 | 33.98 | 25.64 | 14.44 |
> |  | Finetuning | Twitter (77.3k tweets) | Time Separated | 70 | 54.4 | 41.43 | 32.48 | 17.38 |
> | LCBM | Finetuning | Twitter (77.3k tweets) + Youtube (40k) data | Brand Separated | **64.28** | **48.1** | **35.17** | **26.63** | **14.83** |
> |  | Finetuning | Twitter (77.3k tweets) + Youtube (40k) data | Time Separated | **70.23** | **54.54** | **41.52** | **32.54** | **17.45** |
>
>
> The question about detailed information on how the visual encoder handles video input. We apologize for not having a clear enough coverage of this topic. We will make sure to cover this in the next version. The details are:
> The pipeline uses the pre-trained EVA-ViT-G with the Global Multi-Head Relation Aggregator (GMHRA) which is a temporal modeling module.  For token projection, we emply a pretrained QFormer with extra linear projection to incorporate additional query tokens to include video context for training the LLM. While training, the projected video encodings are used to represent video context in the training corpus.
>
>
> (continued in the next comment)

---

> ### Author Response · Authors · 2023-11-16
>
> Regarding (2) (comparison of instruction-tuned models with non-instruction-tuned ones):
> - Prior research has consistently demonstrated the efficacy of in-context learning for larger models. However, our study specifically examines GPT-3.5 and GPT-4 over behavior-related tasks, revealing an incongruity. Notably, the performance variation between 3, 5, and 10 shots is statistically insignificant, indicating a lack of exposure to behavior tokens within these models.
>
> - Furthermore, the stark performance contrast between a 13B behavior finetuned model and significantly larger in-context-finetuned language models (such as GPT-3.5, GPT-4, etc.) is notably high, exceeding 300% (as illustrated in Table 1, 2) and akin to, or even worse than, random performance.
>
> - Importantly, this discrepancy persists even when LCBM is tested in settings disparate from its training data, notably in behavior understanding and behavior domain-adaptation tasks. This highlights the imperative need for LLMs, trained primarily on content tokens (text, vision, videos, etc.), to integrate the knowledge of behavior tokens for comprehensive understanding.
>
>
> References:
>
> - Guilherme Penedo, Quentin Malartic, Daniel Hesslow, Ruxandra Cojocaru, Alessandro Cappelli, Hamza Alobeidli, Baptiste Pannier, Ebtesam Almazrouei, and Julien Launay. The refinedweb dataset for falcon llm: outperforming curated corpora with web data, and web data only. arXiv preprint arXiv:2306.01116, 2023.
>
> - Stella Biderman, Kieran Bicheno, and Leo Gao. Datasheet for the pile. arXiv preprint arXiv:2201.07311, 2022

---

> > ### Comment · Reviewer_73Nf · 2023-11-22
> > **Thank you for the response**
> >
> > Thanks for the author's reply, it solved my problem. I decided to keep the original rating.

---

> > > ### Author Response · Authors · 2023-11-22
> > >
> > > Thank you for acknowledging the rebuttal!
> > > To the best of our knowledge, we were able to answer the two questions you had raised.
> > >
> > > If the questions were addressed, please consider improving the score.

---

### Meta-Review · Area_Chair_bFga · 2023-12-11

**Metareview:**

The submission explores how to train models to optimize for the desired receiver behavior, rather than to just predict symbols. Reviewers appreciated that this is an interesting new take on how to extend the current language modelling paradigm, the strong results compared to GPT3.5 and GPT4, and the new dataset. The main limitations are the somewhat narrow scope of the data (email/Youtube) compared to the very general framing, and the narrow technical contribution. While the paper produced a wide range of scores (not uncommon when trying to do something new), at least some reviewers found it extremely interesting, and I think it will be an exciting addition to ICLR.

**Justification For Why Not Higher Score:**

Probably not one of the top papers at ICLR

**Justification For Why Not Lower Score:**

I think this paper is worth highlighting, and will probably spark some conversations.

---

### Decision · Program_Chairs · 2024-01-16

Accept (spotlight)